# The Rationale for “Laser-Induced Thermal Therapy (LITT) and Intratumoral Cisplatin” Approach for Cancer Treatment

**DOI:** 10.3390/ijms23115934

**Published:** 2022-05-25

**Authors:** Renan Vieira de Brito, Marília Wellichan Mancini, Marcel das Neves Palumbo, Luis Henrique Oliveira de Moraes, Gerson Jhonatan Rodrigues, Onivaldo Cervantes, Joel Avram Sercarz, Marcos Bandiera Paiva

**Affiliations:** 1Department of Otolaryngology and Head and Neck Surgery, Federal University of São Paulo (UNIFESP), Sao Paulo 04023-062, SP, Brazil; renandebrito7@gmail.com (R.V.d.B.); marcelpalumbo@gmail.com (M.d.N.P.); cervantes@uol.com.br (O.C.); 2Biophotonics Department, Institute of Research and Education in the Health Area (NUPEN), Sao Carlos 13562-030, SP, Brazil; marilia.mancini@nupen.com.br; 3Department of Physiological Sciences, Federal University of Sao Carlos (UFSCar), Sao Carlos 13565-905, SP, Brazil; luis.h.o.moraes@gmail.com (L.H.O.d.M.); gerson.ufscar@gmail.com (G.J.R.); 4Department of Head and Neck Surgery, David Geffen School of Medicine, University of California, Los Angeles, CA 90095, USA; jsercarz@gmail.com

**Keywords:** cisplatin, oncological phototherapy, laser-induced thermal therapy, drug resistance, mechanisms of cytotoxicity

## Abstract

Cisplatin is one of the most widely used anticancer drugs in the treatment of various types of solid human cancers, as well as germ cell tumors, sarcomas, and lymphomas. Strong evidence from research has demonstrated higher efficacy of a combination of cisplatin and derivatives, together with hyperthermia and light, in overcoming drug resistance and improving tumoricidal efficacy. It is well known that the antioncogenic potential of CDDP is markedly enhanced by hyperthermia compared to drug treatment alone. However, more recently, accelerators of high energy particles, such as synchrotrons, have been used to produce powerful and monochromatizable radiation to induce an Auger electron cascade in cis-platinum molecules. This is the concept that makes photoactivation of cis-platinum theoretically possible. Both heat and light increase cisplatin anticancer activity via multiple mechanisms, generating DNA lesions by interacting with purine bases in DNA followed by activation of several signal transduction pathways which finally lead to apoptosis. For the past twenty-seven years, our group has developed infrared photo-thermal activation of cisplatin for cancer treatment from bench to bedside. The future development of photoactivatable prodrugs of platinum-based agents injected intratumorally will increase selectivity, lower toxicity and increase efficacy of this important class of antitumor drugs, particularly when treating tumors accessible to laser-based fiber-optic devices, as in head and neck cancer. In this article, the mechanistic rationale of combined intratumor injections of cisplatin and laser-induced thermal therapy (CDDP–LITT) and the clinical application of such minimally invasive treatment for cancer are reviewed.

## 1. Introduction

Cis-[Pt-(NH_3_)_2_Cl_2_], also called cisplatin or CDDP, was the first transition metal complex to be used as a chemotherapeutic agent, and is still one of the most widely used and effective compounds in cancer treatment [1]. It is estimated that 50% of cancer patients will be treated with CDDP at some point [2]. It is considered as a first-line therapy for several types of solid cancer, such as ovarian, testicular, head and neck, and small cell lung cancer [3]. In addition, it is used in combination with other drugs, radiation and immunotherapy for other types of cancer [1].

Cisplatin is composed of the platinum molecule, two relatively inert amine ligands and two labile chlorine ligands in the cis isomeric orientation [3]. The cytotoxic mechanism of CDDP is mainly linked to the formation of DNA adducts through its ability to bind to DNA after undergoing a hydrolysis reaction when entering the cell by replacing chlorine ligands, inducing a series of intracellular signaling pathways that result in apoptosis of the cells [4].

The first reports on cisplatin date from 1845, when the Italian chemist Michele Peyrone first demonstrated the compound cis-[Pt-(NH_3_)_2_Cl_2_], formerly known as the Peyrone salt [5,6]. However, Peyrone did not define the cis isomeric orientation in his studies. In 1893, Alfred Werner, winner of the Nobel Prize in Chemistry for his contributions to stereochemistry, fully defined the chemical structure of cisplatin [7,8]. However, it was only in 1965 that the biophysicist Barnett Rosenberg accidentally (serendipitously) discovered the cytotoxic properties of cisplatin when examining the effects of electromagnetism on the bacterial growth of Escherichia coli strains with platinum electrodes, when he realized that the blocking of bacterial growth was not caused by electromagnetic action, but due to a substance derived from corrosion of the electrodes in the solution, the cisplatin, opening the field for the use of inorganic compounds in cancer treatment [9]. In 1968, Rosenberg started a preclinical study with mice, successfully demonstrating the effectiveness of cisplatin in inhibiting cell growth in tumor cell lines (sarcoma 180 and leukemia L1210) [10]. With the success of preclinical studies, in 1971, the first clinical trial was organized by the National Cancer Institute in the United States. With the undeniable success of cisplatin demonstrated from subsequent clinical trials, cisplatin was approved for the treatment of ovarian and testicular cancer in 1978 by the United States Food and Drug Administration (FDA), and, in 1979, gained approval in the United Kingdom and in several other European countries (Figure 1) [11].

However, despite the great therapeutic success, there are two characteristics of this platinum-based chemotherapy that limit its extensive use in clinical practice: natural and acquired resistance to the drug (which include reduced uptake or increased efflux of cisplatin, increased drug detoxification by cellular thiols, increased DNA repair or tolerance of cisplatin-damaged DNA and the ability to evade cisplatin-induced apoptosis), and serious side effects (mainly ototoxicity, nephrotoxicity, hepatotoxicity, gastrointestinal and neurotoxicity) [12,13,14]. Therefore, the need to develop therapeutic strategies to overcome these problems is highlighted. Several cisplatin analogs have been tested in clinical trials in light of these effects, but only a few have received approval for clinical practice, such as carboplatin, oxaliplatin and nedaplatin in Japan, lobaplatin in China and heptaplatin in Korea [15]. In contrast, the increase in the evidence for photo- and thermoactivity of metal complexes has led to the development and improvement of oncological phototherapy, which includes dynamic phototherapy, experimental photoactivated chemotherapy and photothermal therapy, which are less invasive alternatives aimed at increasing the cytotoxicity of the drug, reducing its side effects and overcoming drug resistance mechanisms [16]. In this context, chemo-photothermal therapy with the application of laser and intratumoral cisplatin appears to be promising due to its photophysical and photochemical properties [17]. In this review, we present the molecular basis that underlies the synergistic effect of the combined approach of laser-induced thermal therapy (LITT) and intratumoral CDDP as a potential, viable and rational approach to cancer treatment.

## 2. Mechanisms of Cytotoxicity of Cisplatin

Cisplatin is dissolved in sterile saline ([Cl^−^] = 154 mM) for intravenous administration in patients. Following the Le Chatelier principle for balance shift, the drug is relatively stable and electrically neutral in the bloodstream, since the plasma chlorine concentration is high (approximately 105 mM), meaning that the labile chlorine ligands of the molecule are replaced quickly [15]. In the bloodstream, about 65–98% of the compound is bound to plasma proteins, especially albumin [18]. One molecule of albumin binds to five cisplatin molecules from different binding sites, with the cisplatin bonding to nitrogen atoms of some amino acids, such as His105, His288, His67 and His247 [19,20]. The last two are related to the transport of zinc, which would explain the zinc disorders that some patients using cisplatin may have; that is, cisplatin occupies the binding site, increasing the amount of serum free zinc [21]. In addition, there is also bonding to sulfidic atoms of the amino acids of Met329, Met298 and Met 548 [19]. As the serum chlorine concentration is slightly lower than the concentration of the solution, there is some degree of imbalance to the right, with the formation of ionic compounds, monohydrolysates, in the bloodstream, which can interact with carbonate and phosphate ions [15].

Currently, it is believed that the main means to achieve cisplatin internalization is by passive diffusion, supported by the observation that cisplatin uptake is linear in accordance with its concentration [22,23,24] and that the accumulation of cisplatin is not inhibited by structural analogues, implying the absence of competition with active transport receivers [25]. Outside the cell, cisplatin is electrically neutral and is not influenced by electrostatic energy from the cell membrane, which keeps hydrophilic ions from crossing it. Upon entering the cell, the formation of cisplatin mono- or dicationic species occurs, requiring high energy (100–300 kJ mol^−1^) for cell evasion by a passive mechanism [26]. A recent study evaluating the rate of cisplatin uptake through passive transport has concluded that it is not possible to exclude the contribution of active transport mechanisms in the uptake of cisplatin; however, it strongly suggests that passive diffusion has an important contribution in this context [27].

Among the active transport mechanisms involved in cisplatin uptake, some receptors have attracted particular interest from the scientific community. The copper carrier protein 1 (Ctr1; SLC31A1) was initially related to cisplatin due to its increased expression in fungi treated with cisplatin [28,29], with further validation from in vitro studies with human tumor cells [30,31]. Subsequent studies confirmed this correlation and suggested that these proteins would be involved with cisplatin uptake [32,33,34,35]. However, increasing evidence points to it being less relevant in this process. This evidence includes observations of the down-regulation of receptors with exposure to cisplatin [36], loss of cis orientation, the extracellular domain rich in methionine and histidine residues, suggesting stable connections, similar to what happens in plasma with albumin, in addition to suggesting that the exchange of oxidative states would be important to enter the cell by this transporter, since copper occurs in two oxidative states, while platinum only in one [15,37]. It has also been suggested that cisplatin is internalized together with Ctr1 through a process of pinocytosis, with the vesicle protecting the interaction with intracellular compounds that inactivate cisplatin [38]. However, it is not clear whether there is an interaction between the cisplatin molecule and the Ctr1 protein [39]. Due to the structural similarity between the Ctr1 and Ctr2 proteins, the effect of the latter on cisplatin uptake was tested [40,41]. However, a recent study using a CRISP-Cas9 genome editing system to eliminate Ctr1 and Ctr2 from HEK-293T cells demonstrated that none of the cell lines exhibited greater sensitivity to cisplatin than the variance in parenteral populations [42].

Other transporters have been linked to the uptake and efflux of cisplatin from the cell. Ionic compound transporters, such as organic cation transporters 1 (OTC-1; SLC22A1) and 2 (OCT-2; SLC22A2) and organic anion transporters 1 (OAT, SLC22A), were mainly related to the toxic effects of oto- and nephrotoxicity [43,44]. The efflux type-P transporters, ATP7A and ATP7B, were related to the efflux of the cisplatin molecules [45], as well as the protein associated with multi-drug resistance (MRP2) and the multi-drug extrusion transporter-1 (MATE1), which were related to the efflux of the cisplatin complex with intracellular proteins rich in cysteine, such as glutathione (GSH) and metallothioneins (MT), which lead to the formation of thiol groups and function as an important mechanism of inactivation (detoxification) of cisplatin within the cell. Decreased uptake, increased efflux and detoxification through these cellular proteins are classically related to cisplatin resistance [46].

Due to the mechanisms of inactivation of intra and extracellular cisplatin, only about 1% of intracellular cisplatin is directed to genomic DNA, the main site of action [4,26]. After entering the cell, the cisplatin molecules undergo a hydrolysis process due to the drastic drop in the concentration of chlorine classically to 4 mM, although studies with direct measurement point to between 12 and 55 mM [26]. The labile chlorine binders are then replaced by one or two water molecules. These compounds are nucleophilic and interact with the genetic material through cisplatin-DNA adducts, mainly in the N7 positions of purine bases [47], with intrastrand cross-links (90–95%—in which 60–65% is for 1,2-d (GpG) and 20–25% is for 1,2-d (ApG)), monoadduct (2%), 1,3-d (GpXpG) intrastrand cross-links (although to a lesser extent; a recent study attributed a contribution to inhibition of transcription and preventing of DNA repair [48]) and interstrand cross-links (2%) [49,50]. These bonds between the cisplatin molecule and DNA alter the molecular structure, preventing DNA replication and transcription. In addition, they activate cell signaling pathways that ultimately result in apoptosis [26].

The remainder, which do not participate in DNA binding, act in other locations, including the cell membrane and mitochondria, through oxidative damage. Cisplatin induces oxidative stress through the formation of reactive oxygen species (ROS), such as hydroxyl radicals, which depend on the concentration and time of exposure to cisplatin [1]. Under physiological conditions, cells control ROS levels by balancing ROS generation with the elimination system (glutathione-GSH, superoxide dismutase-SOD and catalase-CAT). However, in conditions with increased oxidative stress, this regulation is lost. ROSs are responsible for lipid peroxidation, depletion of sulfhydryl groups and alteration of signaling pathways. The mitochondria are the primary target of oxidative stress induced by cisplatin, resulting in loss of sulfhydryl groups of proteins, inhibition of calcium absorption and reduction in the mitochondrial membrane potential, which, associated with Bax (Bcl2 X-linked), results in rupture of the mitochondria [4]. This cleavage releases cytochrome C and pro-caspase-9, which bind to apoptotic protease activation factor 1 and ATP (adenosine triphosphate) to form an apoptotic complex, which activates caspase-9. Activated caspase-9 interacts with a group of caspases (caspase-3, caspase-6 and caspase-7) resulting in apoptosis. This mechanism is also called the intrinsic pathway. Another way of inducing cisplatin-mediated apoptosis is through the cell membrane. The type II transmembrane protein and the Fas ligand (FasL) activate the Fas receptor, which facilitates the formation of the apoptotic complex derived from FADD (Fas-associated death domain), which activates pro-caspase-8. This, after activation, interacts with caspase-3, caspase-6 and caspase-7, leading to apoptosis [1]. The drug accumulation processes, and the formation of cisplatin-DNA adducts are the basis of the synergistic mechanisms associated with cisplatin thermo- and photoactivation.

## 3. Photoactivity and Thermoactivity of Cisplatin

Light or optical radiation can be understood as a representative part of the whole electromagnetic radiation with wavelength within the interval 100 nm to 1000 μm, referring to the ultraviolet (UV), visible (VIS) and IR (infrared) spectra, being responsible for several photochemical processes involved in biological functions, such as the production of vitamin D under UV light, regulation of the circadian rhythm, and hormonal secretion, among many others. However, in addition to the contribution to physiological systems, light has been explored in the diagnosis and treatment of diseases by direct and indirect phototherapy, with therapeutic effects arising from the photoactivation of both endogenous and exogenous compounds. Following the absorption of a photon, a molecule of the drug undergoes one or more primary processes, such as a photochemical reaction of the drug itself, absorption by endogenous molecules, and further energy dissipation decays concerning, for instance, photochemical or photothermal interactions, with energy release and/or electron transfer. The result of the interaction of light with matter depends on many factors, among them on its degree of organization. In this context, metal complexes are useful, since they have a high degree of organization, in addition to a prolonged lifetime [51]. From this, the understanding of these photochemical and photophysical processes has gained particular importance in the development of new approaches to cancer treatment, especially with platinum-based compounds.

Light can interact with biological tissues in several ways, involving three main different photobiologically triggered non-radiative relaxion-based interactions: (a) photochemical, (b) photothermal and (c) photomechanical (three-fold—photoablative, photoplasmic and photodisruprive) [52,53]. The absorption of optical energy E = *hν* = *hc*/λ (J) is needed in order for a tissue to be altered by light (Grotthus–Drapper law). The rate at which a light beam is absorbed by the tissue is given by the irradiance (W/cm^2^) and the tissue absorption coefficient μ_A_(cm^−1^) (considering endogenous contributors or an exogeneous chromophore, such as in PDT) when absorbing a light of wavelength λ. The light parameters involved in different interactions are power density (W/cm^2^) and exposition time (s), and, hence, ultimately, fluence or energy density (J/cm^2^), and continuous wave (cw) or pulsed operations. Photochemical interaction involves photon energy conversion into chemical energy, where excitation triggers chemical reactions, including photoassociation, decomposition, synthesis, activation and isomerization (interaction times cw > 1 s to minutes). Photothermal interaction involves optical energy conversion into heat–thermal energy (e.g., hyperthermia, coagulation, carbonization, vaporization, melting), with lasers being cw, ms or ms pulsed (μs, ms and cw lasers, with interaction times > 1 s to minutes). Photoablative interaction normally involves pulsed UV and produces decomposition of cellular and extracellular components into fragments, resulting in tissue etching (high peak power pulsed lasers 1 ns—1 ms). Photoplasmic interaction involves much higher peak power densities (10^11^ W/cm^2^) in other wavelengths than UV, causing tissues to experience very high electric fields producing photoionization and resulting ablation. Photodisruptive interaction refers to plasma creation through a high electric field responsible for the creation of shock waves (photoplasmic/photodisruptive involves very high peak power lasers with very short pulsed lasers 1 fs—1 ns). Table 1 below represents these light-tissue interaction processes. (In the Table, S represents a singlet state (2S + 1 = 1), T stands for a triplet state (2S + 1 = 3), S is the spin quantum number, * represents an excited state, A, B, C, M are molecular species, and E_kin_ is released kinetic energy).

There are reports of light therapies from ancient Egypt and Greece, with a combination of the ingestion of plants containing psoralen and irradiation by sunlight in the treatment of vitiligo [51]. The term photodynamic therapy was introduced only in 1905 [54]. However, the study of photochemical processes gained prominence in 1912 with questions asked by the chemist Giacomo Ciamician about the potential of light for human civilization in various fields of science, including the following question: “Would it not be advantageous to make better use of radiant energy?” [55]. The concept of coordination compound or metal complex originated 20 years earlier, by the same researcher who determined the molecular structure of cisplatin, Alfred Werner [7]. However, an intersection of the two sciences is relatively new, with the photosensitivity of metallic metals being demonstrated for the first time in 1970 by Balzani [56]. Currently, with the development of oncological phototherapy with platinum-based compounds, we can answer Ciamician’s initial question affirmatively.

Platinum appears as a transition metal of square-planar molecular geometry, i.e., an element whose atom has an incomplete d sublevel or that can become a cation with an incomplete d sublevel, and which are represented in the periodic table by block D. The most important oxidative state of this element is +2, with an electronic configuration d^8^ [57]. Transition metals have photophysical and chemical properties that enable a series of atomic changes arising from specific electronic transitions by energy absorption and energy dissipation, which are involved in actions with anticancer mechanisms, such as photodissociation and/or redox changes, photosensitization and photothermal interaction, which form the basis for photoactivated therapies, such as photodynamic therapy (PDT), photoactivated chemotherapy (PACT) and photothermal therapy [16,58]. In the following, we briefly summarize the inherent properties of the elements in block d, the specific characteristics of platinum, the evidence of cisplatin photoactivity and the main approaches in the context of oncological phototherapy.

### 3.1. Photochemical and Photophysical Properties of Metal Complexes

Szaciłowski et al. pointed out that one of the main advantages of the photochemical activation of transition-metal complexes is the generation of electronic excited states under very moderate reaction conditions [51].

Light can induce electronic transitions of metal coordination complexes between a variety of orbitals, correlated to varied and intense colors provided by metal complexes. Electronic transitions occurring between the metal d orbitals, known as *ligand-field* (LF) or *d-d* transitions, are prohibited by the Laporte selection rule, and hence they occur with very low probability. On the other hand, charge-transfers are strong or high probability related transitions, which take place from orbitals of the ligand to the d orbitals of the metal (*ligand-to-metal-charge-transfer*, LMCT), from the metal d orbitals to the ligand orbitals (*metal-to-ligand charge-transfer*, MLCT), from the d orbitals of the metal to the solvent (*charge-transfer-to-solvent*, CTTS), between orbitals of two ligands without involvement of metal d orbitals (*interligand*, IL), and, finally, between two metal ions of different oxidation states in the same complex *(intervalence charge transfer*, IVCT). Photochemical products are formed according to the photochemical pathways that participate with various photophysical decay processes involving different radioactive and radiationless deactivation decays. More than one type of transition contribution can occur for the same metal complex, yielding complex spectra.

Depending on the wavelength range of the radiation (λ), optical excitation can give rise to various electronic excited state populations, each one displaying distinct reactivities. Hence, convenient excitation wavelengths can tune both photochemical reactivity and routes, e.g., electron transfer mechanisms, involving population of diverse charge transfer states (CT), dissociation, substitution and rearrangement reactions provoked by excited ligand field states (LF), and ligand-centered reactivity due to the population of intraligand states (IL). The effectiveness of a given photochemical process is measured by the quantum yield for the formation of the product of that pathway [51].

The absorption of photonic energy E = *hν* by metal complexes leads to a transition from a ground state (S_1_) to an excited state (S_1_* or S_2_*) through electronic transitions between atomic orbitals. Electronic transitions are classified as allowed and prohibited, and their occurrence probability depends on two main selection rules: (1) the spin selection rule; and (2) the Laporte selection rule [59].
Spin selection rule: ∆S = 0. Transitions with a spin multiplicity change are not allowed, where S = spin quantum number. Transitions between states of the same spin multiplicity (e.g., singlet, triplet) are allowed, and between states of different spin multiplicity (singlet—triplet) are prohibited.Laporte selection rule: ∆ℓ = ± 1 (ℓ = angular momentum quantum number); only transitions between subsequent sublevels are allowed (for example s → p, p → d, d → f …, or p → s, d → p, f → d, …)

The types of electronic transitions involved in the excitation of metals in block d (which includes platinum) can be categorized as [60]:

Metal-centered (MC) transitions (d-d transitions): these are prohibited transitions and typically lead to the occupation of antiligand orbitals, often leading to an elongation of the connection or replacement of the ligand.

Charge-transfer transitions (ligand-to-metal, metal-to-ligand, or to-solvent): permissible bonds which can lead to redox reactions and bond cleavage, reducing the metal center and generating radicals. The production of radicals is a well-documented mechanism of injury to biological structures, such as DNA.

Ligand-centered (LC) transitions or interligand (IL) transitions: permitted connections between ligands.

In general, after absorbing a light photon, the molecule transitions from a ground state to an excited singlet, short-lived state where the excited molecules tend to decay to a lower energy state through non-radioactive processes, by molecular mechanisms such as internal conversion, vibrational relaxation, cooling (quenching); or radioactive processes (fluorescence). On the other hand, an intersystem crossing (ISC) may occur, which leads to an excited triplet state and energy release. This excited triplet state molecule (^3^S_1_) then returns to the ground state (^1^S_o_) through a photon emission, by phosphorescence. The release of energy is related to chemical changes, such as isomerization, dissociation, redox reactions, and substitution. These are among the reactions required in the cisplatin hydrolysis process for the subsequent formation of DNA adducts [51]. In addition, long-lasting excited triplet states may be involved in chemical reactions of type I and type II originating reactive oxygen species, such as in photodynamical reactions.

In this context, the photoactivity of cisplatin (cis-[Pt(NH_3_)_2_Cl_2_]) has already been demonstrated in some studies using different methods. Its ability to undergo photosubstitution reactions when irradiated by light with a wavelength of 350 nm has been reported [61]. In addition, a subsequent study demonstrated the decomposition of cisplatin in a solution containing chlorine induced by sunlight and ultrasound [62]. Studies carried out with a synchrotron, a high-energy particle accelerator that generates powerful, monochromatic electromagnetic radiation, has demonstrated the induction of a cascade of Auger electrons, which are ejected from the valence shell by the energy absorption caused by electronic transitions, demonstrating the theoretical possibility of the photoactivation of cisplatin. Such studies observed a greater number of double-stranded DNA breaks, with repair slower than usual [63,64,65].

### 3.2. Photodynamic Therapy (PDT)

Photodynamic therapy (PDT) is an approach to cancer treatment that involves the use of a photosensitizer and light. It is used to treat cancers, such as lung, superficial gastric, cervical and bladder cancer, as well as head and neck cancer. The mechanisms involved in this approach are oxygen-dependent by type II chemical reactions. The molecule of photosensitizer absorbs the light’s photon and becomes excited through electronic transitions. Hence, oxygen can receive this energy and a transformation occurs from a triplet state (^3^O_2_) to a highly reactive cytotoxic singlet state (^1^O_2_). While the photosensitizer is regenerated, the singlet state oxygen, although short-lived, reacts with cellular compounds, which causes cellular damage and, ultimately, cell death [66]. There are at least three major mechanisms of PDT action: (1) direct cell killing by lethal oxidative damage of tumor cells (necrosis, apoptosis); (2) photodynamic damage of the (neo) vasculature with loss of oxygen and nutrients supply to the tumor (hypoxia and anoxia); and (3) inflammatory and immune responses [67]. However, the requirement for an oxygen environment is a major difficulty as many malignant, and most aggressive, cancer cells are hypoxic [68]. There are several in vitro studies that have demonstrated the synergistic effect of the combination of PDT with cisplatin in the treatment of various types of cancer cells, with better results obtained than for isolated approaches [69,70,71,72,73,74,75]. There have also been some clinical trials probing the efficacy of PDT with concomitant cisplatin [76,77]. Pass, in a phase-3 randomized clinical trial in patients with malignant pleural mesothelioma, compared the standard treatment with surgery, chemotherapy and immunotherapy, associated or not with the use of intratumoral PDT using Photofrin II as a photosensitizer and 630 nm laser light, and found no statistically significant differences in mean survival (14.4 vs. 14.1 months) and median progression-free time (8.5 vs. 7.7 months) [76]. Akopov, in another phase-3 randomized clinical trial on stage IIIA and IIIB central non-small cell lung cancer patients, who were not initially eligible for surgical treatment before neoadjuvant therapy, compared chemotherapy alone with chemotherapy and endobronchial PDT using chlorine E6 as a photosensitizer and 662 nm laser light, before each of the three chemotherapy courses. The arm with PDT therapy was associated with greater resectability with statistical significance (*p* = 0.038), demonstrating that PDT was effective, safe and responsible for the improvement in resectability in this patient population [77].

#### PDT Mechanism

The absorption of the light photon (E = *hν*) by the photosensitizer gives rise to an excited singlet state (S_0_ + *hv* → S_1_). Such excited species have a short lifetime (in the nanosecond range). The excited S_1_ state then decays back to the S_0_ ground state by radiative decay through a singlet–singlet transition (S_1_ → S_0_ + *hv*′), a process known as fluorescence, or to an excited triplet state along with heat release (S_1_ → T_1_ + ΔE) by intersystem crossing (ISC). The T_1_ excited state decays to a singlet ground state (S_0_) by means of a radiative transition, so-called phosphorescence (T_1_ → S_0_ + *hv*″). The lifetime of the T_1_ species (normally in the millisecond range) is sufficiently long for it to interact with the surrounding molecules. Then, the T_1_ state of the excited photosensitizer can undergo two types of reaction [78,79,80,81]:Type-I reaction: involves electron or hydrogen atom transfer, between the photosensitizer in T_1_ state and the biological substrate molecules, producing reactive oxygen species (ROS), which tend to react with ground state ^3^O_2_, resulting in oxidated products;Type-II reaction: refers to the interaction between photosensitizer-excited T_1_ molecules and oxygen in the triplet ground state (^3^O_2_) by energy transfer, resulting in its transition into a highly reactive species, singlet oxygen (^1^O_2_).

Type I and Type II reactions give rise to chemical species capable of oxidizing the tumor cells. As the energy transfer reactions are faster than the electron transfer reactions, mechanism II is generally favored in photo-oxidation reactions. In addition, the main mechanism depends on the photosensitizer type, the excited triplet state yield, the ^1^O_2_ lifetime, the originated ROS stability and properties of the biological medium. Since PDT action is oxygen-dependent, photosensitization cannot occur in anoxic areas.

The light penetration depth into tissue, and thus the PDT action, depends on the light wavelength, which, in turn, must be resonant with the used photosensitizer absorption (normally Q bands), as well as with dosimetric parameters (e.g., laser power, total energy, and fluence (energy density, in J/cm^2^)). Nowadays, photosensitizers absorbing in the deep red and near-infrared wavelengths, with higher absorption coefficients, and/or extinction coefficients, and lower dark toxicity, are preferable to other visible-light-wavelength-absorbed drugs, such as haematophorphyrin-derivative drugs. The extent of photodamage and cytotoxicity is multifactorial and depends on the type of sensitizer, its extracellular and intracellular localization, the total dose administered, the total light dose (energy), the light fluence rate (power density), the oxygen availability, and the time between the administration of the drug and light exposure. All these factors are interdependent. The antitumoral action of PDT is at least three-fold: i) by direct cellular destruction by lipidic peroxidation, ii) through microvasculature destruction, and iii) by inflammatory reactions and immune system response. The triggered inflammatory cascade reactions potentialize antitumoral immunity through leukocyte recruitment and antigen T-cell activation [81].

### 3.3. Photoactivated Chemotherapy (PACT)

Photoactivated chemotherapy (PACT) is another light-based antitumoral approach that utilizes the photochemical and photophysical properties of platinum complexes. This method aims to control where and when a drug is activated, using pro-drugs which are activated upon light radiation, with the major advantage being an oxygen-independent therapy, in contrast to PDT, providing a target therapy with low side-effects to the patient. A pro-drug is a precursor compound that is metabolized or activated in the tissue to generate the active drug. Ideally, a pro-drug should be stable under physiological conditions, being accumulated by cancer cells but not by normal cells, with a long plasma half-life sufficient for the accumulation, while the elimination rate should be rapid enough to avoid toxicity after activation [66]. There are different mechanisms of activation of such compounds, as described below [82]:

Photoreduction: Pt (IV) complexes reduced upon irradiation releasing cytotoxic Pt(II) species and ligands.

Photosubstitution: Pt(II) complexes with photolabile ligand (e.g., curcumin) which undergo ligand dissociation followed by solvent substitution.

Photocleavage of ligand: Photon absorption by the metal center can result in the cleavage of organic bonds, (e.g., C–N, N–O) in photosensitive o-nitrobenzyl alcohol derivatives coordinated to platinum.

Photoswitching: Pt(II) complexes bridged by a diarylethene ligand change the configuration of the ligand upon irradiation to alter the cytotoxic properties.

Although some compounds have undergone clinical trial, none have been approved for clinical use [83]. The great challenge of this approach is to find improved photocytotoxicity by the highest wavelength, providing enhanced light penetration of the tissue [82].

### 3.4. Photothermal Therapy (PTT)

In his monograph, *Hyperthermia and Cancer*, published in 1982, George Hahn described the first experiments on the effects of hyperthermia on localized neoplastic diseases, with the hypothesis that cells in poor nutritional conditions and low pH appeared to be easily inactivated by hyperthermia [84]. Among the possible effects of hyperthermia on neoplastic processes, the combination of chemotherapeutic agents and hyperthermia has attracted interest in the scientific community. One of the first studies describing the synergism between hyperthermia and cisplatin was performed by Meyn (1980), who demonstrated increased cell death of Chinese hamster ovary (CHO) cells with cisplatin action at 43 °C, with an efficiency ten times that at 37 °C, with a cross-linking rate of 6.5, a mechanism usually implicated in increased cytotoxicity, possibly due to cell membrane alterations [85].

Accumulating evidence indicates that heat induction causes ultrastructural changes in tumor cell membranes leading to increased membrane transport of drugs and altered cellular metabolism [86,87]. Higher CDDP uptake after heating also enhances cytotoxicity in proliferating cancer cells and stimulates protein kinases to induce tumor apoptosis [88]. Studies of CDDP binding to DNA have shown that hyperthermia induces more DNA adducts, contributes to increased tumor growth delay, and amplifies interstrand crosslinks with cisplatin [89,90,91,92]. Together, these results provide strong evidence that synergistic effects of heat and cisplatin are likely to improve the therapeutic response of recurrent head and neck tumors following combined drug and laser treatment.

Studies in vitro suggest multiple mechanisms of action for hyperthermia, alone or with CDDP [93,94,95,96,97,98,99,100,101,102,103,104,105,106,107,108,109] (Table 2). The main effects of hyperthermia on cytotoxicity involve conformational alterations on DNA, increase in the intracellular uptake of CDDP, and increase in the number of crosslinks and adducts formed. Hettinga suggested a great increase in the CDDP concentration in resistant cells, with increase in adduct formation and delayed repair of DNA [110]. These mechanisms provide support for clinical trials of simultaneous treatment with CDDP and hyperthermia.

#### Photothermal Interactions

In biological tissues the main chromophores or absorbing molecules include protein, melanin, haemoglobin (oxy and de-oxy), porphyrins, water, and fat, among many others. Distinct chromophores present different absorption coefficients μ_A_(λ) (cm^−1^) to light of the same wavelength (λ), nm. The photon’s absorption from a laser by a tissue chromophore molecule promotes it from its ground state to an excited state (S_0_ + *hν* → S_1_). Decaying from S_1_ back to the ground state may occur radioactively by fluorescence (S_1_ + *hν*′ → S_0_) or non-radioactively to T_1_ by intersystem crossing (ISC) with heat release (ΔE = Q_ISC_). From the excited T1 state, the molecule can return to the original S_0_ ground state radioactively by long-lived phosphorescence. Briefly, in the photothermal interaction, photon absorption leads to further heat dissipation by means of non-radiactive mechanisms (vibrational relaxation (VR), internal conversion (IC), and intersystem crossing (ISC)), giving rise to so-called absorptive-heat (Q). In turn, the transport of the generated absorptive heat inside the tissue is determined by the tissue’s thermal properties (e.g., specific heat, thermal conductivity) and also by its density (also representing its thermal iniertia). The amount of absorptive heat Q is proportional to the absoption coefficient, to the laser power density and the irradiation time.

The possible thermal effects obtained (hyperthermia, coagulation, carbonization and vaporization) depend on the total thermal energy (heat) released and the corresponding finally attained temperature. Whereas hyperthermia and coagulation start to occur from 40 °C and 60 °C, respectively, carbonization takes place above 100 °C and tissue vaporization happens when the irradiated tissue temperature reachs more than 100 °C. Ideally, a high rate of vaporization, along with moderate side thermal damage, are desirable. The intended thermal effect taking place depends on the laser–tissue interaction parameters, such as the tissue’s optical properties, represented by its absorption coefficient μ_A_(cm^−1^), and on the laser parameters, such as the power, power density or irradiance (W/cm^2^), and the exposition time (seconds), total energy, (J). The vaporization rate (μm/s) and the thermal penetration depth δ_TH_ (mm) parameters depend on the abovementioned applied laser irradiation parameters. Photothermal interactions have been used for many decades since the advent of the first medical lasers to produce laser-surgery and laser-induced thermal therapy (LITT) for both coagulation and the ablation of tumors, alone or along with a cisplatin approach protocol.

Thus, photothermal therapy is of great interest for combination with cisplatin or other platinum-based drugs, due to the photothermal energy released in situ, which allows for enhanced toxicity effects accomplished by hyperthermia.

It is hypothesized that hyperthermia increases cisplatin accumulation in part by increasing Ctr1 multimerization and thus greater cisplatin accumulation. Increased Ctr1 multimerization following hyperthermia treatment (41 °C) in vitro, compared to normothermic controls (37 °C), was observed, suggesting that there may be a mechanism for an increased cisplatin uptake in heat-treated cells. Hyperthermia-enhanced cisplatin-mediated cytotoxicity in wild type (WT) cells had a dose-modifying factor (DMF) of 1.8 compared to 1.4 in Ctr1-/- cells because WT cells contained greater levels of platinum compared to Ctr1-/- cells [15,111].

### 3.5. Photobiomodulation

Lasers have been studied and used for cancer treatment and ablation and are widely described in the literature [112,113,114,115]. The thermal ablation of tumors, using a high power laser, is a technique that has been used in oncologic surgery and is named “laser induction thermal therapy,” or LITT [116,117]. Low level laser therapy (LLLT) (or photobiomodulation) involves photochemical interaction and has been used mainly for mucositis treatment in cancer patients [118,119] and for acute radiodermatitis [120].

For tumor ablation, the region that receives the high power laser or LITT suffers irreversible damage and tumor vaporization by high temperature (≥100 °C) [116]. However, the outlying region tissues around receive low level laser without vaporization (40–60 °C), which can have the side effect of the LLLT [116,119,121,122]. Thus, we believe that LITT is able to induce the damage effect and the photobiomodulation induced by LLLT.

In LLLT (or photobiomodulation) therapy, there are three mechanisms of light action on tumor cells. The first is by the direct action of light at high doses, where, at wavelengths of 632 nm, cell death is observed due to possible “overdose” of light [123,124], probably by a stimulus that increases reactive oxygen species. The second is the association of PBM with cytotoxic anti-tumor agents. In this situation, there is a complex mechanism. In tumor cells, the mitochondria undergo a change in their metabolism—instead of performing oxidative phosphorylation, the organelle starts to produce energy through anaerobic glycolysis [125,126]. PBM also acts on cytochrome C oxidase (CcO), an enzyme that corresponds to protein complex IV of the respiratory chain, in addition to being the primary photoreceptor for red light and near-infrared light [127,128,129]. In the stimulation of CcO, the absorbed energy has several effects, such as promoting the upregulation of reactive oxygen species, which assists in cell death and provides enough energy to initiate pro-apoptotic signaling, since apoptosis involves high energy expenditure [130,131]. The third situation is the effect of stimulation of light on the immune system. It was observed that at wavelengths of 660 nm, 800 nm, 970 nm, light is able to reduce the tumor growth mass and increase the recruitment of cells from the immune system, such as T-lymphocytes, and dendritic cells that secrete IFN type I. It also has the ability to reduce the number of highly angiogenic macrophages and to promote the normalization of vessels, which contributes to the control of tumor progression [132].

Several factors have been described which modulate cancer cell proliferation and tumor progression, including (ROS). Low-reactive oxygen species favor the proliferation of tumor cells, while at high levels, they favor the cell apoptosis pathway [133,134,135]. Many cancer cells show an elevation in ROS generation maintaining the oncogenic phenotype, with a redox adaptation by upregulation of anti-apoptotic and antioxidant pathways to promote survival affecting all cancer cells’ behavior [136,137,138]. An imbalance in oxidative stress can modulate cell cycle progression and proliferation, cell morphology, cell motility, energy metabolism, cell survival and apoptosis, cell–cell adhesion, and others [136,137,138,139,140,141]. Intracellular ROS elevation is able to kill cancer cells, while the inhibition of ROS formation can promote apoptosis and cell adhesion in some cancer cells [136,140,141]. To overwhelm the threshold level of toxicity in tumor cells a combination of two or more strategies may be needed to increase ROS levels within these cells [136,137,138,139,140,141].

Oxidative stress can be modulated by antioxidants, pro-oxidants enzymes or molecules, and by laser treatment. In studying the relationship of light with cytotoxic agents (such as cisplatin), we can assess the beneficial association in combating tumor cells by focusing on oxidative stress. There are two signaling pathways for cisplatin, which are called the cytoplasmic and nuclear modules. In the cytoplasm, this agent can tip the redox balance towards oxidative stress, creating DNA damage. However, the drug becomes susceptible to antioxidant enzymes that inactivate its action [142]. In tumor cells, infrared laser can induce higher reactive oxygen species levels and upregulation of ATP production, compared to non-cancer cells [131,143]. In this way, the potentiation of intracellular oxidative stress by combination of cisplatin and laser should be the main mechanism of action to induce the killing of tumor cells, with some selectivity, considering that tumor cells are more sensitive to ROS production compared to non-tumor cells [142].

As mentioned, photobiomodulation can promote molecular changes in the conformation of cytochrome c oxidase in cells with high oxidative stress, increasing ATP production and transient reactive oxygen species. Cisplatin is also able to increase the concentration of ATP and ROS [144,145,146]. High levels of ATP can induce cell death by apoptosis; therefore, the combination of photobiomodulation and cisplatin is favorable to cell death of tumor cells through synergistic action, as verified by in vitro study, with photobiomodulation shown to enhance cisplatin toxicity [70,147].

In the nuclear pathway, lesions induced by the agent generate distortions in DNA that are generally recognized by the cellular repair mechanisms [148]. The nucleotide base excision mechanism removes cisplatin adducts from DNA, preventing its distortion [149]. Another mechanism is related to proteins belonging to the incompatibility repair system, which participates in the resolution of DNA injuries caused by cisplatin [150]. If the damage done to DNA is beyond the cell’s repair capacity, then there is a signal for the programmed cell death pathway [151].

In addition, laser-induced thermal therapy (LITT) enhances the tumoricidal effect of cisplatin as a result of the temperature increase, until a threshold of 43 °C is reached [152]. Increased drug uptake and inhibition of DNA repair or topoisomerase activity at elevated temperatures may also improve tumoricidal effects [66,86,94]. At these temperature levels, heat also appears to reverse acquired resistance to cisplatin [66,86]. Below 43 °C, there is only the effect of reversing the acquired resistance of cisplatin. There is evidence that heat induces ultrastructural changes in cell membranes, resulting in increased susceptibility to cytotoxic drugs and changes in cell metabolism [145].

## 4. Laser-Induced Thermal Therapy (LITT)

Laser-induced thermal therapy (LITT) is a minimally invasive surgical approach based on thermal ablation provided by laser via flexible conductive fibers, acting by external or interstitial radiation. During the last 30 years, LITT has gained attention in various clinical scenarios, such as liver tumors, lung tumors, brain tumors, and recurrent or advanced head and neck tumors, among others. Since its creation in 1983, there have been technical improvements to increase its safety and precision, especially with advances in magnetic resonance (MR)-guided therapy [117]. The basic principles involved include the conversion of light laser energy into photothermal energy (heat) by the absorption of photons by the tissue, as well as thermal diffusion, distributing this photothermal energy progressively at lower levels towards the tissue margins, acting under three mechanisms, as shown in Figure 2: laser-induced coagulation (LIC: > 60 °C), dynamic thermal reaction (RDR: 48–60 °C) and laser-induced hyperthermia (LIHT: 42–47 °C). In the core of the irradiated area, there is virtually instantaneous irreversible cell destruction at temperatures > 60 °C, while the tissue margins may suffer reversible cell damage (42–60 °C), and, in the case of tumors, it becomes a region with a high rate of relapses, acting better in conjunction with chemotherapy [153].

The minimally invasive non-ionizing nature and versatility of the flexible fiber optic method provides some advantages, such as access to places inaccessible by conventional surgery, reduction in unwanted damage to the surrounding tissue (especially if MR-guided), endoscopic access or open surgery, functional improvement (decrease in pain, bleeding, obstruction) with less postoperative morbidity, reduction in hospital stay with minimal anesthesia or even outpatient treatment, generating less physical and emotional impact on the patient and less financial impact on health care providers [117,121,153,154]. Due to these potential benefits, it appears to be a potential and attractive alternative or adjuvant to conventional surgical procedures [117]. The main disadvantages of the method include the limitation of its use for large and irregular tumors adjacent to large vessels, and the inability to precisely control the dispersion, temperature and dose of intended photothermal energy, especially in the interstitial approach [17,155]. Currently, several models of planning and real-time evaluation of the performance of LITT through MR-imaging seek to circumvent these limitations to expand the clinical use of this technique in deep tissues [117].

### 4.1. Physics of Laser Radiation

Laser is an acronym for light amplification by stimulated emission of radiation. Laser radiation is generated through an active medium containing excited electrons, which, under the action of a photon generated by an external source (optical, chemical or electrical), transition from the excitatory state to the ground state, releasing another identical emitted photon, which in turn generates a chain reaction that amplifies the number of photons, ultimately resulting in coherent, collimated, and monochromatic electromagnetic waves with little energy loss [154].

### 4.2. Biological Effects Resulting from the Interaction between Laser Radiation and Tissue

The main biological effect resulting from LITT is thermal damage through the transformation of light energy into photothermal energy [117,155]. The interaction between light energy and tissue, its absorption and distribution, depend on the specific optical characteristics of the laser (wavelength and irradiance) and tissue (absorption coefficient, dispersion coefficient and the anisotropic factor, another parameter that describes the dispersion) [154]. As discussed previously, there are three basic types of light-tissue interactions: photochemical, photomechanical and photothermal interactions [52,53]. In the latter, the most relevant interaction in LITT, the light energy that is absorbed at the atomic and molecular level results in a high energy state, which is exchanged with the environment as heat [117]. Light absorption under irradiation at wavelengths of the near-infrared (NIR) spectral portion has the greatest degree of penetration into tissue and is mainly determined by the amount of water and hemoglobin. In deeper layers of tissue, the light absorption decreases, as postulated by the Lambert–Beer law. The distribution of laser energy in the surrounding tissue basically depends on three processes: absorption, dispersion and bending [154]. The determination of photon propagation, as well as the specific optical characteristics of each tissue, and the cumulative thermal damage and the temperature at each point reached by the radiation, can be estimated by mathematical models from elements of continuous mechanics, thermodynamics, anatomy and physiology, which, if integrated into imaging tests such as MR, enables much more controlled and individualized therapy. This further expands its potential for clinical application in the future which is currently an intense area of research [117].

Ultimately, hyperthermia generated by local heat induces protein and collagen denaturation, enzyme processes, and blood vessel sclerosis, which result in cell death. In addition, from 43 °C, it is already possible to identify coagulative necrosis in the tissue, with the time until cell death is reached depending exponentially on the temperature. However, it is important to point out that photothermal injuries alter the optical properties of the tissue, especially if they result in tissue carbonization (>100 °C), a condition that must be avoided, as most of the light energy is absorbed, light penetration is reduced and impaired and the degree of clotting becomes restricted [154].

### 4.3. Equipment Used for LITT

For LITT, devices with wavelengths in the near-infrared (NIR) spectrum, such as the neodymium laser: yttrium aluminum garnet (Nd:YAG), which operates at a wavelength of 1064 nm and transmits energy with high penetration into tissues (2–10 mm), are particularly used in this context [117,154]. Another type of laser used for LITT are diode lasers, with wavelengths generally varying between 800 and 980 nm (modern devices operate between 600–1300 nm), which have the advantage of the Nd:YAG laser, including a better water absorption coefficient, producing injury more efficiently [154]. Generally, both operate in the range of 2 to 40 W at 110 V and air cooling. The transmission of laser energy can be carried out from optical fibers, produced from silica together with a coating of the same material or similar polymer; or quartz fibers, both of which are capable of transmitting energy over large distances with minimal losses [154]. In the past, the fibers were damaged by excessive heat generated under high power; to overcome these challenges, currently, this problem can be circumvented with an active cooling system using refrigerated catheters [117].

## 5. Laser Photochemotherapy (LPC)

The combination of PDT and cisplatin for cancer treatment presents a unique opportunity to progress research in transition metal complexes as photo-activated chemotherapeutic (PACT) agents and to broaden research into laser photochemotherapy (LPC) [17]. The fields of phototherapy and inorganic chemotherapy both have long histories, while inorganic photoactivated chemotherapy (PACT) offers both temporal and spatial control over drug activation and has remarkable potential for the treatment of cancer [59].

Anthracyline derivatives, such as adriamycin and daunomycin, are the most common anticancer agents that interact with light to elicit fluorescence, membrane photo labeling, laser activation and killing of tumor cells [156,157]. Studies with these anticancer agents include reports of excited states of these drugs and the further generation of radical oxygen species which appear to be wavelength dependent, from 313 to 498 nm [157]. The cytotoxicity of several anthracycline derivatives is significantly enhanced by continuous wave green light of argon (514 nm) or KTP (532 nm) laser illumination of different types of cancer, both in vitro [158,159] and in vivo [157,160,161,162]. Minton and Ketcham [163] first described enhanced potentiation of the oncolytic capability of a pulsed ruby laser when combined with cyclophosphamide in a melanoma tumor model in 1965. Carmichael et al. [164] and Li and Chignelli [158] raised the possibility of using chemotherapeutic drugs to photosensitize tumors since many are chromophoric and absorb light at specific wavelengths of the visible spectrum leading to energy transfer and oxygen species which cause photo-oxidation. Based on this concept, several clinical studies combining chemotherapy with laser therapy were undertaken in the mid-1990s [165,166]. Previous studies with anthracyclines, such as adriamycin and daunomycin, have shown that tumor cell peroxidation and membrane photo-labeling occur immediately after drug uptake [167,168]. Both drugs cause chemical bonding to many different cell membrane proteins [158]. Anthracyclines initially bind to tumor cell membranes before transport in the nucleus and are considered to be poor type-II photosensitizers, which do not generate significant amounts of singlet oxygen [168,169,170]. Nevertheless, these chemotherapy agents are the most tested in the hope of enhancing three distinct anti-cancer effects when combined with visible light: phototoxicity, thermal-toxicity and chemotherapy per se [159]. Recent attempts to reduce anthracycline toxicity have led to the development of a variety of anthrapyrazole derivatives, including DUP-941 (CI-941), which possess reduced side-effects but which have been shown to retain clinical efficacy in breast cancer patients [171]. An unusual property of DUP-941 is that it has an over 100-fold increased photooxidation potential compared to anthracyclines, as reported by Reszka et al. [172].

Based on the strong evidence of the premise above, our group at the University of California Los Angeles pioneered a combined treatment protocol of photodynamic therapy (PDT) and laser photochemotherapy (LPC) for treating a patient with an unresectable large recurrent nasopharyngeal squamous cell carcinoma [161,168]. After two sessions of LPC and three of PDT, delivered over a three-month period, complete tumor regression was observed at the level of the nasopharynx and the patient remained free of local recurrence for six months. This case illustrates the potential benefits of these two adjunct modalities for palliative treatment of head and neck and other tumors and a change in paradigm for these patients, since this approach CDDP–LITT may be repeatedly applied, extending survival with quality of life, focusing on maximally delaying the time to recurrence as fundamental tactics for advanced head and neck cancer evolution. Therefore, the proposed treatment is designed to transform cancer into a manageable chronic disease. This initial result was also a stepping stone for us to proceed with our research program in laser chemotherapy from “bench to bedside” funded by the National Cancer Institute from 1993 to 2016 [115].

In addition to the advantage of photo-activation, several of these light sensitive chemotherapeutic agents have been reported to exhibit enhanced toxicity in tumor cells after photothermal-activation [113,114,162,173,174]. As a result, a potentially useful and less invasive approach for treatment of malignancies is to combine imaging-guided interstitial laser surgery with conventional chemotherapy [159,175].

Several FDA-approved anti-cancer drugs are highly photosensitive or heat-responsive, including anthracycline derivatives and cisplatin [113,152]. This experimental technique broadens the concept of tumor ablation based on the thermal denaturation of malignant cells to include anti-cancer drug activation by laser energy [157,171,176]. In 1995, our group at the University of California Los Angeles (UCLA) [159] demonstrated that intratumor injection of an anthracycline derivative, combined with interstitial laser fiberoptic photoactivation, was a feasible means of enhancing photodynamic therapy treatment, as recently shown by van Veen et al. [177]. Hence, laser photochemotherapy explores two distinct mechanism of antitumor action: (1) direct toxic effects, and (2) additional photochemical and/or photothermal toxicity [177,178,179,180,181]. These drugs may be injected intravenously at concentrations lower than normal chemotherapeutic levels, or at higher intratumor doses, reducing systemic toxicity, while enhancing local tumoricidal effects by laser photoactivation in situ [17,181]. With the supporting evidence of translational studies, photochemotherapy has been established as an alternative treatment for retinoblastoma [182]. Most of these studies were conducted in children where there are several standardized clinical protocols, in particular for unilateral retinoblastoma [183,184,185].

Because head and neck cancers are accessible for surgery and have well-described loco-regional biological behavior, they are an ideal model to test combined laser energy delivered via interstitial fiberoptics and chemotherapeutic agents activated by photothermal energy as an alternative, as a less invasive treatment for cancer [116,120]. Long-term remission and tumor eradication may be possible by combining intratumor chemotherapy with photothermal energy delivered via laser fiberoptics. In this model, cisplatin and hyperthermia have been shown to be an effective combined therapy in the laboratory and in recent clinical trials. In addition to therapeutic benefits and improved tumoricidal effects, combining intralesional anti-cancer drugs with interstitial photo-thermal laser treatment reduces systemic toxicity and is less invasive than conventional chemotherapy or surgical resection [116,152,175].

## 6. Initial Testing of Laser-Induced Thermal Therapy and Chemotherapy as an Alternative Treatment for Head and Neck Cancer

The treatment of head and neck cancer has evolved dramatically. Advances in surgery, radiation therapy, and chemotherapy have improved locoregional control, survival, and quality of life [15,97,110]. However, despite significant improvements in treatment, head and neck cancer remains a source of considerable morbidity and mortality, as a substantial proportion of patients will succumb to their disease [115,121,186]. Some studies estimate that 20% of patients would qualify for palliative care at the time of initial diagnosis, with an average survival of five months within this cohort [186,187]. Therefore, despite important advances in current therapy with surgery, radiation, and chemotherapy, nearly one half of all head and neck cancer patients will develop persistent or recurrent disease [188,189]. Most recurrent head and neck cancers and their locoregional metastases are accessible through laser fiberoptics and serve as an interesting disease model to test intratumor chemotherapy with laser energy delivery via interstitial fiberoptics using drugs activated by photochemical and photothermal energy as a less invasive treatment alternative for cancer [152,175,176,177,190,191]. Moreover, there has been no generally accepted standard of care for recurrent head and neck cancer [192,193,194]. Phase I and II clinical trials have been conducted using laser-induced thermal therapy (LITT) in a stepwise fashion to palliate patients with advanced and recurrent head and neck tumors as an alternative to more radical and at times disabling surgery [195,196,197]. Our team has treated over 500 patients with recurrent head and neck cancer using LITT, rendering UCLA one of the most experienced medical centers in the country for using Nd:YAG laser therapy in the palliative treatment of recurrent head and neck cancer [197].

Thermal ablation of malignant tumors by infrared light energy emission by an Nd:YAG (neodymium: yttrium-aluminum garnet) laser has developed into a multidisciplinary subspecialty in oncologic surgery presently named “laser-induction thermal therapy”, or LITT [117,198]. Near infrared (NIR) light penetrates human tissues with limited depth, thereby providing a method to safely deliver non-ionizing radiation to well-defined target tissue volumes [199]. LITT is a standardized procedure, where the deployment of fiber optics is guided by imaging or direct visualization, and has some promising results in palliative care for unresectable tumors of the central nervous system, gastrointestinal system, breast, liver and prostate [122,200,201,202]. LITT has received growing acceptance as an alternative palliative, minimally invasive therapy for recurrent and/or advanced head and neck cancer patients who do not respond to conventional treatment [115,121].

## 7. The Development of the Combination of Cisplatin and Laser Treatment for Cancer

Initial work on chemotherapy and lasers was reported in clinical models investigating combined palliative therapy for inoperable esophageal cancers. Semler et al. [203] noticed significant improvement in quality of life in 21 of 24 patients with advanced gastrointestinal tumors treated by systemic chemotherapy administered before intraluminal endoscopic Nd:YAG laser thermal ablation. Survival benefits of this alternative combined treatment for palliation of advanced upper gastrointestinal tumors were also reported by Mache et al. [204]. Mason [205] confirmed these findings when reporting on patients with esophageal cancer that presented a significant reduction in need for additional laser therapy to maintain swallowing when adjunctive chemotherapy was given before laser treatments. This was an outstanding study based on a randomized clinical trial, confirming that patients with esophageal cancer presented a significant reduction in need for additional laser therapy to maintain swallowing when adjunctive chemotherapy with epirubicin and cisplatin was given before laser treatments. Firusian [206] compared different anti-cancer agents with Nd:YAG laser for endoscopically treated patients with advanced stenotic upper gastrointestinal cancer. Best median survival (8 months) was observed in a group of 13 patients who received combined endovenous cisplatin in combination with other anticancer drugs. The author also reported that combined drug and laser therapy was more effective in preventing rapid cell proliferation at the tumor margins. Thirteen patients with stenotic upper gastrointestinal cancers, treated by endoscopic recanalization with laser ablation and systemic cisplatin in combination with other anticancer drugs, responded more favorably compared to laser alone as a palliative approach for advanced malignant disease. In most of the 13 patients, combined therapy led to immediate patency of the upper alimentary tract and 8 months median survival in the cohort studied. In this study it was also found that local laser thermal effects were enhanced by systemic chemotherapy leading to additional ablation of malignant cells down to 7–8 mm depth compared to 4 mm for laser treatment alone. Thus, it was concluded that combined drug and laser therapy was more effective in preventing rapid cell proliferation at the tumor margins [206]. Expanding the same concept, Vogl et al. [207] have proposed a combination of chemoembolization and LITT for liver tumors with promising results. All this previous work was undertaken in Europe, and only in 2000 did our group report their first clinical experience with this form of combined therapy in eight patients with recurrent head and neck tumors conducted in the United States testing systemic chemotherapy (CDDP at 80 mg/M^2^) followed 24 h later by palliative Nd:YAG laser thermal ablation [116]. Four of the eight patients treated in this manner remained alive after a median follow up of 12 months. A total of twelve tumor sites were treated, and complete responses were seen in the following anatomic locations: oral cavity (*n* = 3), oropharynx (*n* = 1), hypopharynx (*n* = 1), and maxillary sinus (*n* = 1). The median survival for these patients was 9.5 months. The adverse effects of treatment included mild alopecia in an 82-year-old female and a bout of gastrointestinal infection in another patient [116]. A total of 21 patients were treated in this study that showed minimal toxicity of the combined treatment. However, the therapeutic benefit was not significant because of the great variability of the tumor sites analyzed [208]. One of the patients treated in this series presented with a recurrent SCCA of the neck after having previously undergone a reconstructive free flap transfer [209]. The patient underwent six concurrent treatment sessions using the protocol mentioned above and demonstrated an unusually long period of survival (i.e., over five years). The remarkable survival of this patient suggests that the combination of LITT and chemotherapy warrants further investigation as an alternative treatment for patients with recurrent head and neck cancer. Further studies have shown that even more effective eradication of head and neck cancer is possible by combining LITT with local intratumor injections of CDDP [210]. These studies led to the first case published in the literature combining intratumor injection of cisplatin and Nd:YAG laser by our group [175].

## 8. Intratumor Injections of Cisplatin and Laser-Induced Thermal Therapy (CDDP–LITT): A Change in Paradigm for Advanced Head and Neck Cancer

Initial experimental studies combining intratumor injections of cisplatin followed by local hyperthermia were carried out by Kitamura et al. [211] in a melanoma model. The authors demonstrated that the combined treatment led to a six-fold decreased tumor growth rate of melanoma and improved prognosis without nephrotoxicity. In the mid-1990s, several studies explored local adjunct chemotherapy to eliminate marginal tumor regrowth and improve final outcomes after surgery and radiation [113,114,152,174,175,176]. These studies had a strong rationale based on two relevant reports by authors Begg et al. [210] and Theon et al. [212] who proposed adjunct local chemotherapy combined with tumor resection as a more effective approach for cancer treatment. The local chemotherapy proposed was based on a therapeutic implant of cisplatin in a gel vehicle (CDDP/gel) which consisted of purified bovine collagen (a protein carrier), cisplatin and a vaso-constrictor, epinephrine [213]. This therapeutic implant provided sustained release of the drug in tumors and greatly reduced systemic toxicity [214]. Recent studies with human SCCA transplant models combining intratumor chemotherapy with LITT encourage further development of this novel combined therapy to successfully eradicate marginal disease for treatment of recurrent head and neck cancer [112].

An important contribution to CDDP–LITT treatments was a study by Kanekal et al. [215] reporting that a 5 min interval between intra-tumor injection of CDDP (in solution) and LITT would retain a significant amount of the anti-cancer agent in the tumor margins that would benefit from the synergistic association with laser thermal therapy. These additional experiments by our group in 2008 were central in defining our treatment strategy for combined intratumor injections of cisplatin in solution (1 mg/cm^3^ of tumor) followed five minutes later by laser-induced thermal therapy using the Nd:YAG laser @50Watts (CDDP–LITT). Establishing this clinical protocol for CDDP–LITT led to Institutional Review Board approval at the University of California Los Angeles (UCLA—IRB 013-000152) and the Ethics in Research Committee at the Federal University of São Paulo (UNIFESP—CEP 00-40 2008) in Brazil, as a co-participant in an initial nephrotoxicity study in nine patients described by Palumbo et al. [115] in 2017. A total of 22 patients were treated in this toxicity study confirming the feasibility and compliance of CDDP–LITT within Brazil’s National Public Health System (Sistema Único de Saúde [SUS]) operational procedures [216]. Eighteen men and four women (median age 62.5 years) were enrolled. Thirty-six treatments were performed, and the average procedure time was 82 min. Twelve patients underwent one procedure, seven had two, two had three, and another patient had four. The average time interval between treatments was 5.6 weeks. The average survival was 8.6 months (range: 4–32.1 months), which doubled the survival for this specific patient population in the city of São Paulo, as shown in a recent study by Amar et al. [217].

The collaboration with UNIFESP has been a crucial step to expand our laser chemotherapy research program, given that advanced head and neck cancers (AHNC) have a particularly unfavorable prognosis in Latin America, where the five-year survival rate is lower than 30% for squamous cell carcinomas when diagnosed at stage IV [218,219]. For instance, in a study of this specific patient population at UNIFESP, Eugenio [220] reported that 73.2% of patients were Stage IV when first diagnosed by a specialist. The time delay in diagnosis for these patients frequently led to clinical upstaging that influenced the prognosis and limited median survival to 3.8 months [221]. In this scenario Brazil’s health care system faces numerous challenges caring for patients with advanced head and neck cancer: inadequate funding and inequitable distribution of resources and services [222]. Consequently, an increasing number of cancer patients will need more aggressive multimodality therapy that is more costly [222]. Therefore, we visualize an important role for CDDP–LITT as an affordable, outpatient treatment for advanced/recurrent head and neck cancer.

## 9. Initial Studies on Microvascular Collapse (MVC) Enhancement by CDDP–LITT

The rationale for this approach is that, during laser therapy, high photothermal laser energy levels are delivered to the area of maximum obstruction in the tumor core of advanced esophageal cancer inducing irreversible coagulative changes and lower levels at the tumor margin [206]. Less energy is delivered to the margins because of the higher risk of organ perforation or damage to neighboring tissues. Infrared Nd:YAG laser (1064 nm) applied at 50 W, continuous wave, showed that laser thermal therapy for debulking procedures causes boiling of tissue water (^3^100 °C) and subsequent irreversible thermal damage, which ultimately leads to photo-evaporation at the tumor core as described previously [199]. Local recurrence is related to reversible cellular thermal damage on the margins treated with sub-therapeutic energy levels (40–60 °C) in head and neck tumor ablation and other organs and systems [115,117,121,122]. Enhanced laser-induced thermal therapy (LITT) by adjuvant intratumor chemotherapy using cisplatin (CDDP) in the region of sub-therapeutic laser energy levels for improved cancer cells eradication was first described in a murine model with human SCCA transplants by Paiva et al. [152].

Since 1992, our group has been exploring laser photochemotherapy (LPC) using monochromatic light to enhance the “killing” threshold in tumors containing light and/or heat-sensitive anticancer agents [161]. In this sense, we have demonstrated that LPC application with intratumor injected anthracyclines or platin derivatives has been a useful treatment model for marginal tumor re-growth after laser thermal ablation of advanced/recurrent head and neck (SCCA) squamous cell carcinoma [113,152]. Solid tumors have several potential barriers to drug delivery that may limit drug penetration, such as alteration in the distribution of blood vessels, blood flow, interstitial pressure, and microcirculation in the tumor [223,224]. Therefore, intratumor injections of chemotherapy attaining high systemic levels of the cytotoxic drug often cause systemic toxicity without reaching effective concentrations in the tumor [225]. However, cisplatin is also associated with adverse traits, such as nephrotoxicity and drug resistance [226,227]. Cisplatin nephrotoxicity is cumulative, dose-dependent, and is frequently featured as an acute renal failure (20 to 30%), hypomagnesaemia (40%), Fanconi anemia-like syndrome, renal tubular acidosis, hypocalcaemia, increased excretion of sodium, renal concentration defect, proteinuria, erythropoietin deficiency, microangiopathy and chronic renal failure [227]. Doses above 50 mg/m^2^ cause a decrease in glomerular filtration rate and must be minimized to improve treatment tolerability of combined treatment when associated with radiation, hyperthermia or with other drugs, which have been widely used in the last two decades [228]. Cisplatin ototoxicity is dose-dependent and cumulative, and is associated with reduced quality of life, increased incidence of depression, social isolation, and neurocognitive and psychosocial delay in children. The accumulation of cisplatin in inner ear cells (in hair cells, supporting cells, stria vascularis and nerve cells) results in increased formation of reactive oxygen species and decreased antioxidant enzymes (glutathione peroxidase, superoxide dismutase, catalase and glutathione reductase), which through lipid peroxidation, damage to nucleic acids and oxidative modifications of cellular proteins, result in cell death and, consequently, sensorineural hearing loss, usually of high frequency, bilateral, irreversible, associated or not with tinnitus, which may be transient or permanent. The duration, the number of cycles administered, the method of administration, the presence of chronic kidney disease and genetic factors, such as polymorphisms in genes involved in antioxidant action, influence the degree of hearing loss [229]. Heat increases both susceptibility to cell cytotoxicity to cisplatin and intracellular drug absorption, which amplifies the drug effectiveness with less systemic toxic effect [230].

The first authors to raise the possibility of a microvascular barrier (collapse) to drug extravasation during intratumor chemotherapy concomitant with hyperthermia were Mizuuchi et al. [231] in 1996. The authors demonstrated, in human transplanted tumors, that simultaneously combining hyperthermia with carboplatin led to greater intratumor retention of carboplatin and prolonged tumor doubling growth time, possibly due to some microvascular disarrangement. The authors’ findings suggest that these microenvironment changes led to higher retention of carboplatin that potentiated the anti-cancer drug’s activity and its synergistic interaction with hyperthermia-enhancing cytotoxicity [231]. Additional studies at Harvard University have consistently demonstrated the sub-therapeutic temperatures in the margins and the importance of coagulation necrosis due to microvascular collapse in the tumor periphery using radio frequency (RF) in different animal tumor models and histology [232,233]. The investigators also observed a five-fold increase in intratumoral uptake of liposomal doxorubicin in tumors treated with RF compared with tumors that received the drug alone, with preferential chemotherapy accumulation in the “hyperemic zone” surrounding the thermally ablated area [234,235]. These hyperemic zone changes to the subtherapeutic margins (periablational area) were recently identified as microvascular collapse (MVC) promoting greater retention of anti-cancer agents, that was never reported in RF or laser thermal treatment alone [236].

With regards to the “periablational area” (MVC prone areas) in LITT applications for patients, we developed a preliminary translational phase II study in 1998 that reproduced the sub-therapeutic (40–60 °C) thermal distribution modeling in the tumor margins proposed by Marchesini et al. [199] (infra-red 1064 nm; Nd:YAG laser irradiating a coherent light at continuous mode @50Watts; energy density: 2200–3300 J/cm^2^) [193,233,237]. The same lasing parameters were also simulated for recurrent oral cavity tumors, and were recently cited by You et al. [238] and Heyman [239] as a feasible translational model for laser photochemotherapy (LPC) or photoactivated chemotherapy (PACT) with conventional anti-cancer drugs in a variety of release systems, including direct intratumor drug injections for cancer treatment [214,240,241]. In the past 10 years, these laser parameters have been the standard settings used by our group when performing LCT using cisplatin in head and neck cancer patients [116,121,209]. Platinum-based chemotherapy has been the standard for head and neck cancer, and one of the advantages using the settings proposed is that CDDP is heat-activated in the range of 43–270 °C [242,243]. Probably the most successful platform for combined drug and laser treatment was reported by Wang et al. [244] in 2001. The investigators observed 100% complete response in 31 patients (25–91 months follow-up) with T2 esophageal tumors using Nd:YAG laser and intra-tumor 5-fluorouracil and mitomycin. A crucial study using autoradiograms of mice bearing squamous cell carcinoma SCCA tumors demonstrated a significant advantage of intratumor injections of cisplatin in solution over cisplatin as an implant, which led to the translational application of CDDP–LITT in humans [223].

Initial results in three patients treated with intratumor injections of cisplatin (1 mg/cm^3^ of tumor) and laser-induced thermal therapy (CDDP–LITT) were reported in 2012 by Ribeiro et al. [245] demonstrating preliminary data on the impact of microvascular collapse in the outcome of treatment that was well-accepted and toxicity non-existent.

None of the early nephrotoxicity markers (BUN, creatinine, magnesium, type I urine and proteinuria at 24 h) tested on the patients showed changes at these levels of CDDP intratumor injections [115]. Findings on these preliminary reports show that absence of local or systemic adverse effects is the proof-of-concept that MVC plays a significant role in CDDP–LITT for cancer treatment. Based on this proof of concept, our group will move on to a traditional Phase I study escalating doses starting at 75 mg CDDP/cm^3^ of tumor to better understand the role of MVC and potentially increase positive outcomes of PCT as a minimally invasive treatment for cancer. These studies were already approved by the ruling IRB committees.

In theory, the peripheral vascular collapse of arteries and veins produced by LITT will isolate blood flow to and from the tumor, thereby impeding drug (CDDP/sol) washout to the rest of the body [223]. Consequently, high cisplatin concentration in the tumor will potentiate the cytotoxic synergistic combination of drug and heat, and promote more effective treatment [224]. Celikoglu et al. [246] reported that intratumor injections of cisplatin may be increased up to 40 mg in a clinical trial testing local chemotherapy (CDDP/sol) and radiation for inoperable bronchogenic tumor, where no adverse effects were observed. As for the development of CDDP–LITT, this initial experience indicates both safety and therapeutic potential for the palliation of advanced head and neck cancer. However, safety and feasibility must be confirmed by longer follow-up and further dose escalation (>75 mg CDDP/cm^3^) in a Phase I formal study to determine máximum-tolerated dose (MTD) and demonstrate tangible benefits for patients.

Therefore, considering, (1) the evidence of improvement in tumor cytotoxicity provided by the photoactivation and thermoactivation of CDDP [69,70,71,72,73,74,75,76,77,93,94,95,96,97,98,99,100,101,102,103,104,105,106,107,108,109], (2) the evidence of increased cellular uptake of cisplatin under hyperthermia by the tumor, especially by increasing Ctr1 multimerization [5,109,111], (3) local microvascular collapse (MVC) acting as a barrier to drug dissemination to systemic circulation with higher tumor concentration of cisplatin [231,232,233,234,235,236], and (4) Phase I clinical studies demonstrating low incidence of systemic side effects [115], we believe that the CDDP–LITT approach increases local cytotoxicity with a lower incidence of recurrence and possible reduction of drug side-effects in patients who are not candidates for conventional surgical therapy. Further investigation by phase III clinical trials is required for external validation of these data.

## 10. Future Directions

The curiosity, imagination and persistence of Professor Barnett Rosenberg led to the discovery and development of the globally important anti-tumor drug, cisplatin. Cisplatin is an established and effective treatment for recurrent and metastatic head and neck squamous cell carcinoma/M HNSCC, and many studies have investigated cisplatin treatment in combination with other agents. However, though major progress has been made in surgery, radiation, and chemotherapy for the treatment of malignancy during the last 20 years, there has been little improvement in the survival of patients with recurrent or advanced head and neck cancer. Even when treated with first-line therapy (cisplatin + 5-fluorouracil + cetuximab), overall survival is only 6–10 months, indicating a further need for novel chemotherapeutics and treatment regimens. Because of their ease and accessibility for surgery and loco-regional biological behavior, head and neck cancers serve as an ideal model to test the combination of laser energy delivered via interstitial fiber-optics and chemotherapeutic agents activated by photothermal energy as an alternative, less invasive treatment for cancer.

Phototherapies, including photodynamic therapy (PDT) and laser-induced thermal therapy (LITT), have proven to be effective treatments for certain cancers. Nevertheless, PDT has been slow in becoming a mainstream cancer therapy for solid tumors, possibly due to treatment complexity and the challenges of establishing optimum treatment parameters (such as the drug and light dose or the drug to light interval). Laser-induced thermal therapy (LITT) is a standardized procedure, where the deployment of fiber optics is guided by imaging or direct visualization and has some promising results in palliative care for unresectable tumors of the central nervous system, gastrointestinal system, breast, liver and prostate. LITT has received growing acceptance as an alternative palliative, minimally invasive therapy for recurrent or advanced head and neck cancer patients who do not respond to conventional treatments. A large number of preclinical and clinical studies have demonstrated that the combination of PDT with conventional chemotherapeutics (e.g., doxorubicin, cisplatin) are more effective than monotherapies. The impact of these synergistic effects is not only a direct sum of the damages caused by both modalities, but also includes the effects on tumor vasculature, and, in some cases, the induction of an immune response. However, the rationale and understanding of treatment procedure, such as whether chemotherapeutics should be administered before or after irradiation, are still not well understood.

Thirty years ago, our group pioneered laser photochemotherapy application in human subjects with recurrent head and neck cancer using a monochromatic light delivered via interstitial fiber-optics to enhance the “killing” threshold in tumors containing light and/or heat-sensitive anti-cancer agents (adriamycin and cisplatin). Additional studies in a syngeneic human tumor model showed resounding evidence that marginal injection of cisplatin (CDDP) improved laser-induced thermal therapy (LITT) outcomes significantly. This was based on the fact that, during LITT, high doses of thermal energy are directed to the tumor core, leading to irreversible damage and vaporization of tumor cells (100°). In contrast, lower doses directed to outlying regions (margins of 40–60 °C) result in reversible changes to the cells of the tumor margin, with higher risk of recurrence in these locations. Tumor regrowth in the margins was deterred by local cisplatin injections (CDDP) combined with laser-induced thermal therapy, which is the basis for the development of CDDP–LITT from bench to bedside.

Since 1996, the clinical application of cisplatin in CDDP–LITT for palliation of recurrent head and neck cancer has evolved from systemic (IV), local implant (gel) to intratumor injections of CDDP in solution, which is currently used by our team. In 2012, we described the important role of microvascular collapse in the tumor periphery during combined laser thermal therapy and local chemotherapy, which was recently confirmed by other researchers as well. This led to the first study in nephrotoxicity by our group, sanctioning the safety and technical feasibility of high intratumor doses of cisplatin for more effective CDDP–LITT treatment. Therefore, one can envision CDDP–LITT being utilized in tandem with adjuvant biological therapies as a stepwise process that can be repeated as needed, and possibly even transforming cancer into a manageable chronic disease. This research will benefit from recent advances in the development of low-cost imaging systems for real-time monitoring during minimally invasive procedures, which is the quintessential goal of treatment for recurrent head and neck cancer developed by investigators at UCLA.

## Figures and Tables

**Figure 1 ijms-23-05934-f001:**
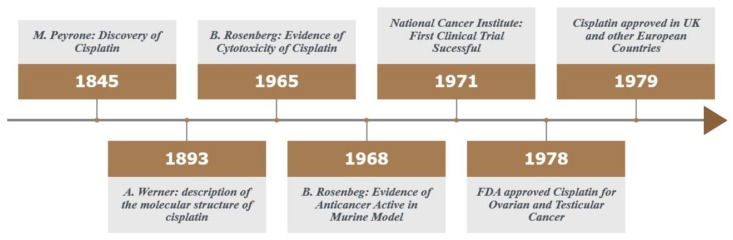
The timeline of cisplatin approval in United States and Europe.

**Figure 2 ijms-23-05934-f002:**
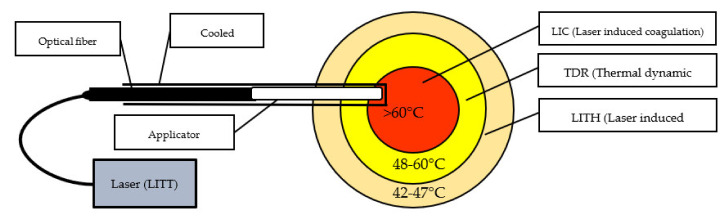
Graphic representation of photothermal mechanisms in laser-induced thermal therapy (LITT).

**Table 1 ijms-23-05934-t001:** Non-radiative relaxion pathways (taken from Boudoux, 2016). * and ** refers to excited and double-excited states, respectively.

Process	Representation
**Absorption**	
From ground state	S + *hν* → S *
From an excited state	S * + *hν* → S **
**Radiative processes *(energy dissipation by photo-re-emission)***	
Fluorescence	S * + *hν*′ → S
Fosforescence	T * + *hν*″ → S
**Non-radiative processes *(energy dissipation without photo-re-emission)***	
**Photochemical effects**	
Photoassociation	A * + B → AB *
Photodecomposition	A * → B + C
Photoisomerization	A * → A′
Electron transfer	A * + B → A^+^ + B^−^, A^−^ + B^+^
Energy transfer	A * + B → A + B *
**Photothermal effects**	
Intersystem crossing (ISC)	S * → T *
Internal conversion (IC)—vibrational relaxation	S * → S
Collision induced relaxion	S * + M → S + M
**Photoablative effects**	
Excitation	(AB) → (AB) *
Dissociation	(AB) * → A + B + E_kin_
**Photoplasmic effects**	
Ionization	A * → A^+^ + e^−^
Photodisruptive	
Photodisruption	A + shockwave → B + C

**Table 2 ijms-23-05934-t002:** In-vitro findings of the interaction of cisplatin and hyperthermia.

Author	Cell Lineage	Findings
**Wallner, K. (1986)** [93]	Chinese hamster ovary cells (CHO) (in vitro)	Dose enhancement ratios increased from 1.4 to 6.5 over the temperature range of 39–43 °C. Cellular accumulation of platinum at 37 °C in the sensitive cells was 2.3- to 3.3-fold greater than that in the drug-resistant cells. Cellular accumulation of DDP was increased by factors of 1.5 and 2.2 at elevated temperature.
**Herman, T. (1988)** [94]	Squamous cell carcinoma CDDP-sensitive (SCC-25) (in vitro)	The dose-dependent cytotoxicity of 1-h exposures to CDDP was markedly increased at 42 °C and 43 °C in comparison to 37 °C, and this effect was of the same magnitude in both cell lines (enhancements of approximately 1.5 logs at 42 °C and 2.5 logs at 43 °C).
	Squamous cell carcinoma CDPP-resistance (SCC-25/CP) (in vitro)	
**Herman, T. (1988)** [95]	EMT6 cells (in vitro)	In EMT6 cells, the cell killing enhanced 2 decades by 10 µM CDDP at 42 °C compared to the cell killing at 37 °C. In Lewis lung carcinoma growing in the legs of C57 mice, there was an increase of about 2.5-fold in the tumor growth delay produced by CDDP with the addition of heat.
	Lewis lung carcinoma (in vivo)	
**Herman, T. (1989)** [96]	EMT6 cells (in vitro)	There were approximately 2 decades enhancement in cell killing by 10 pM CDDP at 42 °C compared to 37 °C. At 42 °C, CDDP was able to gradually alter the gel electrophoretic mobility of the plasmid DNA to near that of the linear form. This change also occurred at 37 °C but at a much slower rate.
**Calabro, A. (1989)** [97]	**18 biopsy specimens (in vitro)**	*Experimental conditions were adopted to simulate “therapeutic” trials: (a) temperature of 37.0 °C, 40.5 °C or 42.5 °C; (b) hyperthermic duration of 30, 60, or I20 min.* - A significant decrease (*p* < 0.0001) in the IC_90_ value was observed in 39 (42%) of the 92 heat-CDDP combinations tested in 16 tumors. - The 40.5 °C hyperthermia significantly decreased (*p* < 0.02) the IC_90_, in 33% (15 of 46) of heat-CDDP combinations; significantly decreased (*p* < 0.0007) the CDDP IC_90_ in 52% of cases (24 of 46) at 42.5 °C. - The 60- and 120-min exposures to hyperthermia plus CDDP were more effective than the normothermic CDDP treatment in 41% (13 of 32) (*p* < 0.04) and 56% (18 of 32) (*p* < 0.003) of cases, respectively. *If the corrected IC_90_ was still significantly lower or higher than the IC, observed at 37.0 °C, the interaction was defined as synergistic or antagonistic, respectively.* - Heat-CDDP combinations were significantly more synergistic (*p* < 0.001), decreasing the IC_90_ in 37% (34 of 92) of combinations.
	Sarcoma human cell line (7 specimens)	
	Colon carcinoma human cell line (3 specimens)	
	Ovarian carcinoma human cell line (2 specimens)	
	Lung carcinoma human cell line (1 specimen)	
	Carcinoid human cell line (1 specimen)	
	Breast carcinoma human cell line (2 specimens)	
	Melanoma human cell line (2 specimens)	
**Cohen, J. (1989)** [98]	JM, a human acute lymphoblastic leukemia T-cell (in vitro)	*TER: the slope of the survival curve at an elevated temperature divided by the slope of the curve at 37.0 °C; The slope of the survival curve for cisplatin alone at 37.0 °C is arbitrarily taken to be −1.00*. - Cisplatin killing: TER was 2.6 at 40.5 °C and 3.6 at 41.8 °C. - Survival of JM cells: TER was 2.6 at 41.8 °C.
**Zaffaroni, N. (1989)** [99]	Human cutaneous or lymph nodal malignant melanoma cell (in vitro)	Synergy between heat and CDDP was observed in 7% of cases treated with the lowest drug dose and 38% of cases treated with the highest (40.5 °C), with only a slight increase in the frequency of synergy at 42 °C.
**Los, G. (1991)** [100]	CC531 carcinoma inoculated intraperitoneally in WAG/Rij rat	*In vivo, rats were treated intraperitoneally with cisplatin (5 mg/kg) in combination with regional hyperthermia of the abdomen (41.5 °C, 1 h)*. - Enhanced platinum concentrations were found in peritoneal turnouts (factor 4.1) and kidney, liver, spleen and lung (all around a factor 2.0) after combined cisplatin–hyperthermia treatment. - The thermal enhancement ratio (TER) using lethality as endpoint was 1.8.
**Majima, H. (1992)** [101]	Chinese hamster ovary cells (CHO) (in vitro)	Pre-heating at 43 °C enhanced cDDP cytotoxicity given immediately after heating, decreasing this enhancement within 24 h to an additive level.
**Yano, T. (1993)** [102]	Transplantable human esophageal cancer (ESO-2) in nude mice (in vivo)	- The combination of 4 mg/kg of CDDP and 43 °C heating for 30 min effectively depressed tumour growth in comparison with the individual treatment. - The mean relative tumour weight of the combination group at 3 weeks after the treatment was 15% of that of the control group without treatment. - Pre-heating at 42 °C for 30 min did not influence the inhibition of tumour growth by CDDP alone or the concentration of CDDP in tumour. When pre-heating at 42 °C for 30 min was performed at 6 or 12 h prior to the combined treatments of 2 mg/kg of CDDP and 43 °C hyperthermia for 30 min, however, tumour growth depression by CDDP-hyperthermia was diminished.
**Takahashi, I. (1993)** [103]	EMT6/KU cells (mouse mammary tumor cells) (in vitro)	- The cytotoxicity of CDDP was enhanced at 43 °C, within 90% cytotoxic concentration (IC_90_) was reduced 2.9-fold. - When exposed to IC_90_ drug concentration at 43 °C for 2 h simultaneously, the intracellular platinum concentration increased 1.9-fold for CDDP
**Kubota, N. (1993)** [104]	HMV-I human malignant melanoma cells (in vitro)	For cell survival, the thermal enhancement ratio was 3.38 for cDDP at 44 °C for 30 min.
**Ohno, S. (1994)** [105]	Leukemia L1210 cells (in vitro)	- Simultaneous treatment with heat (41.5 °C, 60 min) and cisplatin produced maximal cell killing with a 4-fold decrease in the 50% growth-inhibitory concentration (IC_50_) of the platinum complex. Super-additive cell killing was also shown when cells were exposed to heat before cisplatin treatment, whereas no thermal enhancement in cisplatin-mediated cytotoxicity was observed in cells given heat after exposure to cisplatin. - 2- to 3-fold increase in ISC formation was observed in cells given heat before or during cisplatin exposure, whereas heat after cisplatin treatment did not alter either the formation or the reversal of ISC compared with cisplatin alone.
**Kusomoto, T. (1995)** [106]	FSaII murine fibrosarcoma cells (in vitro and in vivo)	- Greater than additive killing of FSaII cells with CDDP and hyperthermia occurred only if the drug and heat exposures were overlapping or simultaneous. - Platinum levels in the cells were determined after 1-h exposure of the cells to a concentration of the platinum complexes required to kill 90% (1 log) of the cells at 37 °C. There was a 4-fold increase in platinum in the cells when CDDP (5 µM) and heat exposure was simultaneous; this corresponded to a 2.5-1og increase in cell killing.
**Ohtsubo, T. (1996)** [107]	Human pharyngeal carcinoma (in vitro)	Simultaneous or post-hyperthermic CDDP treatment for high-hyperthermia (above 42.5 °C) and simultaneous CDDP treatment for low-hyperthermia (below 42.5 °C) were the most effective means of CDDP thermochemotherapy with hyperthermia.
**Ohtsubo, T. (1997)** [108]	Human maxillary carcinoma (in vitro) - CDDP-sensitive (IMC-3) - CDDP-resistance (IMC-3-DDP)	Heating at 40 °C potentiated CDDP cytotoxicity in both cells, with thermal enhancement ratios (TER) of 1.48 (IMC-3) and 1.94 (IMC-3-DDP), and enhanced the platinum accumulation by factors of 1.4 (IMC-3) and 1.8 (IMC-3-DDP).
**Ohtsubo, T. (1997)** [109]	Human pharyngeal carcinoma KB cells	There was a significant increase in CDDP uptake after hyperthermia at 44 °C.

## Data Availability

Not applicable.

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
