# Peer review of "The Rationale for “Laser-Induced Thermal Therapy (LITT) and Intratumoral Cisplatin” Approach for Cancer Treatment"

_ijms, 2022, doi:10.3390/ijms23115934_

Round 1

Reviewer 1 Report

General comments:

This review summarizes the mechanistic rationale of combined intratumor injections of cisplatin and laser-induced thermal therapy (CDDP-LITT) and the clinical application of such minimal invasive treatment for cancer.

Major comments:

  1. The basic information for LITT is not enough. Please provide more basic information for LITT. Moreover, the main difference between X-ray and LITT may need to describe.
  2. The limitation of LITT application may need to mention.

Minor comments:

Line 54: (seripendituously)à change to the “non-bold” type

Line 235: Table 1: Non-radiatiive à Table 1: Non-radiative

Author Response

The authors take the opportunity to thank the reviewers for their valuable comments, questions and suggestions. We have responded to all questions, as discussed below. The comments and suggestions were relevant, and we have updated the manuscript accordingly. If any further modifications are required kindly let us know.

Reviewer 2 Report

Manuscript No. ijms-17155444

„The Rationale for “Laser-induced Thermal Therapy (LITT) and Intratumoral Cisplatin” Approach for Cancer Treatment” for International Journal of Molecular Sciences

Comments:

  1. Lines 202-203. One of the terms "in this context" should be removed from the sentence
  2. Line 290. The word 'stationary' is crossed out. It should be deleted.
  3. Please standardize the wavelength notation: Compare: Lines 357 and 363.
  4. Line 531 and in the rest of the text please correct the abbreviation of the enzyme cytochrome c oxidase with CcO. The current acronym (COX) suggests cyclooxygenase.
  5. Lines 538-540. What the authors mean 'the recruitment of cells from the immune system, having action on T lymphocytes, and dendritic cells', which are cells of the immune system influencing other immune cells? Unless it was the cells of the immune system such as cytotoxic T lymphocytes, dendritic cells or NK cells. Please clearly state it in this sentence.
  6. I am asking the Authors for a short explanation in the text of what would be the reason that the side effects of cisplatin in LITT or PDT may be reduced? I mean to present the cellular or molecular basis of the potential reduction in the nuisance of the classic side effects of CDDP.
  7. Line 852. Please mention and briefly describe in the text the effect of CDDP ototoxicity, as was done for the nephrotoxicity of this cytostatic.

Author Response

(The authors gave the same response as above.)
